# A New Trick of Old Dogs: Can Kappa Opioid Receptor Antagonist Properties of Antidepressants Assist in Treating Treatment-Resistant Depression (TRD)?

**DOI:** 10.3390/ph18020208

**Published:** 2025-02-03

**Authors:** Shaul Schreiber, Lee Keidan, Chaim G. Pick

**Affiliations:** 1Department of Psychiatry, Tel Aviv Sourasky Medical Center, Tel Aviv 6423906, Israel; shaulsch@tlvmc.gov.il; 2Dr. Miriam and Sheldon G. Adelson Clinic for Drug Abuse Treatment and Research, Tel Aviv Sourasky Medical Center, Tel Aviv 6423906, Israel; 3Faculty of Medicine & Health Sciences, Sagol School of Neuroscience, Tel Aviv University, Tel Aviv 6905904, Israel; 4Sylvan Adams Sports Institute, Tel Aviv University, Tel Aviv 6905904, Israel; leekeidan@mail.tau.ac.il; 5Department of Anatomy and Anthropology, Faculty of Medicine & Health Sciences, Tel Aviv University, Tel Aviv 6905904, Israel; 6Dr. Miriam and Sheldon G. Adelson Center for the Biology of Addictive Diseases, Tel-Aviv University, Tel-Aviv 6905904, Israel

**Keywords:** treatment-resistant depression, opioids, mice, antinociception, mianserin, mirtazapine, venlafaxine, reboxetine, risperidone, amisulpride, zolpidem

## Abstract

**Background/Objectives:** Approximately one in five individuals will experience major depressive disorder (MDD), and 30% exhibit resistance to standard antidepressant treatments, resulting in a diagnosis of treatment-resistant depression (TRD). Historically, opium was used effectively to treat depression; however, when other medications were introduced, its use was discontinued due to addiction and other hazards. Recently, kappa opioid receptor (KOR) antagonism has been proposed as a potential mechanism for treating TRD. The main research question is whether commonly used psychotropic medications possess KOR antagonist properties and whether this characteristic could contribute to their efficacy in TRD. **Methods:** We investigated the antinociceptive effects of many psychotropic medications and their interactions with the opioid system. Mice were tested with a hotplate or tail-flick after being injected with different doses of these agents. **Results:** The antidepressants mianserin and mirtazapine (separately) induced dose-dependent antinociception, each yielding a biphasic dose–response curve. Similarly, the antidepressant venlafaxine produced a potent effect and reboxetine produced a weak effect. The antipsychotics risperidone and amisulpride exhibited a dose-dependent antinociceptive effect. The sedative–hypnotic zolpidem induced a weak bi-phasic dose-dependent antinociceptive effect. All seven psychotropic medications elicited antinociception, which was reversed by the non-selective opiate antagonist naloxone and, separately, by the kappa-selective antagonist Nor-BNI. **Conclusions:** Clinical studies are mandatory to establish the potential efficacy of augmentation of the treatment with antidepressants with these drugs in persons with treatment-resistant depression and the optimal dosage of medications prescribed. We suggest a possible beneficial effect of antidepressants with kappa antagonistic properties.

## 1. Introduction

While the term ‘depression’ is frequently used in common language to describe the reaction of an individual to an unpleasant or saddening event, major depressive disorder (MDD) is a severe, debilitating mental disorder with a lifetime risk of mortality due to suicide [1,2]. It is estimated that close to one in five people would experience an episode of MDD and that, at any given time, it affects approximately 350 million people worldwide and is one of the leading causes of mental disability [3]. It can present at any age across the lifespan, and it is, in a way, a kind of “basket diagnosis”, as notable differences in biological vulnerability, age of onset, risk factors, symptomatic presentation, and comorbidities are present among people with the same diagnosis. MDD is, hence, a very heterogeneous disorder. Importantly, approximately 30% of people with MDD are resistant to conventional treatments based on antidepressant medications that target monoaminergic (serotonin, noradrenaline, and dopamine) receptors [4].

There are several definitions for ‘treatment-resistant depression’ (TRD), indicating that the definition of TRD currently lacks consensus [5]; it is commonly defined as a failure of two subsequent treatment trials that have been adequate in terms of duration and dose [6]. TRD patients have significantly more outpatient visits and are at least twice as likely to be hospitalized [7,8] and to commit suicide [9]. TRD is, therefore, a costly disease associated with the extensive use of depression-related and general medical services.

Once the commonly prescribed antidepressant medications nowadays (the selective serotonin reuptake inhibitors (SSRIs), the selective noradrenaline serotonin reuptake inhibitors SNRIs, and the multimodal serotonin stimulator) fail to achieve the goal of remission, various augmentations are available (i.e., lithium, triiodothyronine (T3), some antipsychotics at low dose) with or without psychotherapy [10]. However, TRD may resist all those pharmacological strategies, necessitating looking for new directions and old/historic remedies that may prove beneficial. One focus of research is on rapid-acting hallucinogens, such as psilocybin, which was marketed in the 1960s as medicine by Sandoz (now Novartis) as a “promoter” for people with TRD, but some years later was made a schedule 1 drug and was banned for clinical and research [10,11,12]. Another research focus is ketamine [13,14], for which a nasal spray formulation (esketamine) has been developed and approved for use in medical settings only. However, the potential addictive properties of these substances are of concern, and limiting their administration to medical settings only adds further burden on the accessibility and economics of treatment [15].

Another focus of the research is based on historical experience with opium which for centuries, was largely used as a treatment for ‘melancholia’ (the old term for depression) and for ‘refractory melancholia’ (nowadays referred to as ‘TRD’) [16,17,18,19]. However, opiates and opioids are highly addictive (even more than the hallucinogens mentioned above), and their diversion to illegal manufacturing and smuggling has brought about a severe opioid crisis in the USA and other Western countries [20,21,22].

To date, several types of opioid receptors have been discovered: the m (mu) receptor (MOR), k (kappa) receptor (KOR), d (delta) receptor (DOR), and nociceptin/orphanin FQ receptor (NOR), while other types have been proposed (e.g., s (sigma), e (epsilon), and z (zeta) opioid receptors are being studied). Within these different types there are subsets of subtypes (i.e., m1, m2, k1, etc.) [23]; however, a detailed review of the opioid system is far beyond the scope of this paper.

The main research question is whether some commonly used psychotropic medications possess KOR antagonist properties and, if so, whether this characteristic could contribute to their potential efficacy in TRD. Based on a large scale of basic studies (mouse models of antinociception) of the opioid interactions of various psychotropic medications (antidepressants, antipsychotics, and hypnotics), we have suggested a possible beneficial effect of those antidepressants with agonistic interactions with the opioid system [24]. Following the interesting observations on the possible antidepressant effect of substances exerting k (kappa) receptor antagonist properties [25], such as the partial agonist and partial antagonist buprenorphine [26] found to alleviate depression in subjects with opioid use disorder (OUD) on maintenance treatment with buprenorphine [27,28], as well as reports on its ability, at a low dose, to augment antidepressants’ effects [29], some new potential antidepressants are investigated (e.g., Navacaprant) [30]. Some of these medications are supposed to be non-addictive, as they are antagonists of an opioid receptor with no agonist properties implicated in opioid-related abuse [30].

We are now suggesting to focus on the k (kappa) opioid antagonistic properties of the long series of psychotropic medications already studied and reported for their interactions with the opioid system [31,32,33,34,35,36,37], aiming to suggest their possible use for TRD, as they are already FDA-approved for the treatment of depression or associated mental conditions (e.g., psychotic depression or sleep disorders) [31,32,33,34,35,36,37].

## 2. Results

### 2.1. Dose–Response Curves

#### 2.1.1. Mianserin Antinociceptive Effect

We evaluated the antinociceptive effect of mianserin on mice using a hotplate assay. At doses from 1 to 25 mg/kg, mianserin administered i.p. produced an antinociceptive effect in the hotplate test in a dose-dependent manner (Figure 1). The antinociceptive effect observed with 1 mg/kg was 12%, while its effect with 25 mg/kg mianserin raised to 72%. As the mianserin dose increased beyond 30 mg/kg, the hotplate latencies declined to baseline, yielding a biphasic (inverse U shape) dose–response curve.

#### 2.1.2. Mirtazapine Antinociceptive Effect

Screening of mirtazapine in mice demonstrated its efficacy as an antinociceptive agent in the hotplate assay. Therefore, we evaluated the antinociceptive effect of mirtazapine on mice in this analgesic assay. At doses from 1 to 7.5 mg/kg, mirtazapine-administered i.p. produced an antinociceptive effect in the hotplate test in a dose-dependent manner. The antinociceptive effect observed with 1 mg/kg was 20%, while at 7.5 mg/kg, it rose to 70% as the mirtazapine dose increased. When the mirtazapine dose increased beyond 10 mg/kg, the hotplate latencies declined to baseline, producing a biphasic dose–response curve (inverse U shape) (Figure 2).

#### 2.1.3. Venlafaxine Antinociceptive Effect

Venlafaxine induced a dose-dependent analgesic effect following i.p. administration (Figure 3). The ED_50_ for mice in the hotplate assay for venlafaxine was 46.7 mg/kg.

#### 2.1.4. Reboxetine Antinociceptive Effect

Reboxetine induced a weak antinociceptive effect (Figure 4). At 10 mg/kg, it reached its maximal effect of 30% analgesia.

#### 2.1.5. Risperidone Antinociceptive Effect

Risperidone induced a potent and dose-dependent antinociceptive effect following i.p. administration (Figure 5). The ED_50_ for mice in the tail-flick assay for risperidone was 26.4 mg/kg.

#### 2.1.6. Amisulpride Antinociceptive Effect

Amisulpride induced a dose-dependent antinociceptive effect following s.c. administration. The ED_50_ for mice in the tail-flick assay for amisulpride was 36.6 mg/kg (Figure 6).

#### 2.1.7. Zolpidem Antinociceptive Effect

The evaluation of zolpidem in the hotplate analgesic assay in mice was performed. The groups of mice (n ≥ 15) were injected with various doses of zolpidem. Zolpidem yielded a biphasic dose–response curve: At 15 to 80 mg/kg, zolpidem administered s.c. induced an antinociceptive effect in the hotplate test in a dose-dependent manner (Figure 7). The antinociceptive effect observed with 15 mg/kg was 10%; with 80 mg/kg, it elevated to 80%. As the zolpidem dose increased beyond 90 mg/kg, hotplate latencies declined.

### 2.2. The Sensitivity of Antidepressant Drugs to the Antinociceptive Effect of the Opioid Receptor Antagonists

All the antidepressant drugs in the present study were sensitive to the antinociceptive effect of both opioid receptor antagonists’ studies (the non-selective naloxone and the kappa-selective Nor-BNI).

### 2.3. The Sensitivity of Antidepressant Drugs to the Antinociceptive Effect of the Kappa Opioid Receptor Agonist U50,488H

Mianserin-U50,488H (κ subtype) interactions: We gave the selective agonists of the κ subtype U50,488H with or without an inactive dose of mianserin (0.5 mg, i.p. Table 1). We found an almost 10-fold shift to the left in the dose–response curve (*p* < 0.05). The mechanismED_50_ of U50,488H without mianserin was 4.8 mg/kg, s.c.; with mianserin, it was 0.5 mg/kg, s.c. These results suggest that mianserin, when administered with κ opioid receptor subtypes, significantly potentiates antinociception mechanisms.

Mirtazapine-U50,488H (κ subtype agonist) interactions. We gave the selective agonists of the κ subtype U50,488H, with or without an inactive dose of mirtazapine (0.25 mg, i.p. Table 1). No differences were found between the groups. The ED_50_ of U50,488H without mirtazapine was 4.4 mg/kg, s.c., and with mirtazapine, it was 3.1 mg/kg, s.c. These results suggest that mirtazapine, when administered together with κ opioid receptor subtypes, does not alter nociception.

Venlafaxine-U50,488H (κ subtype) interactions: The selective agonist of the κ subtype U50,488H was injected with or without an inactive dose of venlafaxine (2.5 mg/kg, i.p. Table 1). An almost six-fold shift to the right in the dose–response curve (*p* < 0.05) was obtained. The ED_50_ of U50,488H without venlafaxine was 5.7 mg/kg, s.c., and with venlafaxine, it was 1.0 mg/kg, s.c. These results suggest that venlafaxine, when administered together with opiates, significantly potentiates antinociception mediated by κ receptor subtypes.

Reboxetine-U50,488H (κ subtype) interactions: We did not check the interaction as we found reboxetine to possess a weak antinociceptive effect (mediated by nonselective opioid receptors).

Risperidone-U50,488H (κ subtype) interactions: The selective agonist of the k-one subtype U50,488H was injected with or without an inactive dose of risperidone (5 mg/kg, i.p., Table 1). A 15-fold shift to the left in the dose–response curve (*p* < 0.05) was obtained. ED_50_ of U50,488H without risperidone was 4.4 mg/kg, s.c., and with risperidone was 0.3 mg/kg, s.c.

Amisulpride-U50,488H (κ subtype) interactions: Coadministration of amisulpride with U50,488H did not change k-induced antinociception. The ED_50_ of U50,488H without Amisulpride was 1.9 mg/kg, s.c., with Amisulpride, it was 2.2 mg/kg, s.c. We found no significant shift in the dose–response curves, indicating no considerable augmentation of the antinociceptive effects of opioid receptor agonists by amisulpride (Table 1).

Zolpidem-U50,488H (κ subtype) interactions: The coadministration of zolpidem with U50,488H did not potentiate k-induced antinociception, with statistical significance. No significant differences were found between the dose-dependent curves with and without zolpidem.

## 3. Discussion

Opioid receptors are distributed widely throughout the brain, with varying concentrations in different regions. Areas with high opioid receptor density include the cortex, nucleus accumbens, thalamus, hypothalamus, amygdala, periaqueductal gray, locus coeruleus, rostral ventromedial medulla, and the spinal cord (especially the dorsal horn) [23]. The opioid system interacts with other receptors in the CNS (e.g., the gabaergic receptors), and its involvement in various clinical conditions is possible.

In a series of previous studies, we assessed the antinociceptive effects of several psychotropic medications (antidepressants, antipsychotics, and hypnotics) in a mouse model of acute pain. The aim was not to quantify the analgesic potency of the medications but to assess the possible interactions of these medications with the opioid system. We found several of them exerted antinociceptive effects involving the opioid system [24]. Others had no or only weak opioid properties involved in their antinociceptive effects [31,34,35,37,38,39,40], while others yet had no antinociceptive effect at all [40,41,42].

In the present study, we focused on the k (kappa) opioid antagonistic effects of some medications, a property under extensive research nowadays, while searching for new possible treatments for depression, particularly treatment-resistant depression (TRD). All seven psychotropic drugs in the present study were sensitive to the antinociceptive effect of both opioid receptor antagonists studied (the non-selective naloxone and the kappa-selective Nor-BNI).

The tetracyclic antidepressant mianserin exerted an inverse U-shaped dose–response curve of antinociception (indicating a “therapeutic window” effect), reversed by the non-specific opiate antagonist naloxone, implying the mediation of this effect through the opioid system. This effect was reversed by naloxone and (separately) by Nor-BNI. When U50,488H (κ subtype opioid agonist) was added to mianserin, it significantly potentiated the antinociception mechanisms.

The antidepressant mirtazapine (a piperazinoazepine, not related to any known class of psychotropic drugs), exerted a biphasic dose–response curve (an inverse U-shaped) dose–response curve of antinociception (indicating a “therapeutic window” effect), reversed by the non-specific opiate antagonist naloxone, implying the mediation of this effect through the opioid system and by the kappa antagonist Nor-BNI, implying involvement of the kappa opioid receptors. When U50,488H (κ subtype opioid agonist) was added to mirtazapine, no augmentation of the antinociceptive effect was seen, implying no additional involvement of the kappa opioid agonist in mirtazapine’s antinociception.

The serotonin–noradrenaline selective reuptake inhibitor (SNRI) antidepressant venlafaxine induced a dose-dependent antinociceptive effect, reversed by naloxone and (separately) by Nor-BNI. When the kappa agonist U50,488H was injected with or without an inactive dose of venlafaxine, results suggested that it significantly potentiated antinociception mediated by κ receptor subtypes. The noradrenaline selective antidepressant reboxetine induced only a weak antinociceptive effect, antagonized by naloxone and Nor-BNI; hence, we did not check the interaction with the kappa agonist U50,488H.

The antipsychotic neuroleptic risperidone (a benzisoxazole derivative, one of the “atypical” neuroleptics) induced a potent, dose-dependent antinociceptive effect following i.p. administration antagonized by naloxone and by Nor-BNI. When injected with the selective agonist of the k-one subtype U50,488H, a significant augmentation effect was evident.

Amisulpride induced a dose-dependent antinociceptive effect, antagonized by naloxone and, separately, by Nor-BNI. Co-administration of amisulpride with the κ opioid subtype agonist U50,488H did not yield any change, concluding that there is no considerable augmentation of the antinociceptive effects of opioid receptor agonists by amisulpride.

Zolpidem (a non-benzodiazepine hypnotic drug, one of the “z-compounds” group) induced a weak bi-phasic (inverted V-shape) dose-dependent antinociceptive effect (indicating a “therapeutic window” effect), antagonized by the nonselective naloxone and by all opioid receptor subtypes antagonists (m1- and m2-opioid receptor antagonist b-FNA, the selective m1-opioid receptor antagonist naloxonazine, the selective d-opioid receptor antagonist naltrindole, the k1-opioid receptor antagonist Nor-BNI). No significant effect was noted when the selective k agonist U50,488H was co-administered with zolpidem (see Table 2).

The potential roles of opioid receptors in motivation and major depressive disorder have been known for decades [43], and indeed, until the late 1940s, opium was a recognized treatment for depression, in general, and (what nowadays is called) ‘treatment-resistant depression’ [16]. However, opioids are highly addictive, and their use for the treatment of depression is not an option once the hazards are known. The exceptions are persons with opioid use disorder (OUD) maintained on methadone or buprenorphine, where findings are that some of the persons with depression benefit more when treated with buprenorphine, as it treats both their OUD and their depression [17,28,29,44]. This is attributed to the kappa opioid receptor antagonism properties of the buprenorphine for a review of the kappa opioid receptor and its new directions for the treatment of pain, anxiety, depression, and drug abuse, see Khan et al. 2022 [45].

This effect may also manifest while augmenting the therapeutic effects of conational antidepressants with the oral nonselective opiate antagonist naltrexone [46].

The potential antidepressant properties of kappa antagonists are a focus of attention for pharma research while trying to develop new directions in antidepressant medications outside of the ‘me-too’ innumerable available compounds that act on the serotonin and noradrenaline paradigm of depression (with or without lesser effects on other neurotransmitters thought to be involved in depression, i.e., the phenylethylamine venlafaxine, which in vitro, blocks mainly the synaptosomal uptake of serotonin, to a lesser extent that of noradrenaline, and, to a notably lesser degree, of dopamine [47]. The antidepressant effects of buprenorphine are thought to derive from its kappa antagonist properties, and several such compounds are studied. Fava et al. reported in 2020 on a successful (though a small sample) proof of concept clinical trial of a kappa-selective opioid receptor antagonist augmentation in TRD [48]. Another kappa opioid receptor antagonist studied (aticaprant) was reported by Demyttenaere et al. in 2024 to restore ‘interest and pleasure’ when added to conventional SSRI or SNRI treatment (not achieved before the augmentation) [49], and again, in another study by Hampsey et al. [50].

While electroconvulsive therapy (ECT) has proven to be the most efficacious treatment for depression (including its most severe manifestation—the ‘psychotic depression’), the controversies and stigma that accompany it prevent its use in many cases, as needed [51,52]. Other productive technologies for the treatment of TRD available during the last 20 years include rapid transcranial magnetic stimulation (rTMS) [53,54,55], found to be ‘as good as’ ECT (but not in the case of psychotic depression), vagus nerve stimulation (VNS) [56], and lately—still under research—the deep brain stimulation (DBS) [50,57,58]. On the pharmacological side, various formulations of esketamine (nasal spray, s.c. injections) have been developed [58,59], and psilocybin and other psychedelics are under research [12].

Treatment-resistant depression is a severe form of depression associated with high intensity of symptoms and markedly increased mortality due to suicides and accidental overdoses. In a recent population study, the researchers identified all individuals with a diagnosis of major depression (MDD) who were treated with an antidepressant aged 15 to 65 years during 2004–2016 in Finland [9]. Persons with over two treatment trials were defined to have TRD. Their cohort was large (176,942 individuals with MDD), with typical characteristics (63% women, the median age at index diagnosis 40 years, 11 % fulfilled the TRD criteria, and were followed-up for a median of 8.9 years). In the TRD, there were 6.1 deaths/1000 person–years and 5.6/1000 person–years in the non-TRD.

All-cause mortality was 17% higher in TRD compared to non-TRD. In TRD, increased mortality was observed for suicides and accidental poisonings but not for natural causes. A higher proportion of accident drug overdoses was observed in TRD than in non-TRD (62 % vs. 42 %, respectively) [9].

The need for an efficacious treatment approach to TRD is clear, and the burden of the currently available modalities (that impose ambulatory administration of treatment by a health care person, be it an rTMS technician, a nurse for the esketamine injection, or a psychotherapist during the psilocybin assisted therapy) on the striving health systems is impeding the possibility of expanding treatment as needed. Hence, the need for an approach that can easily be implemented at home (medication taken by the patient) is evident. One such treatment approach would be the endorsement of low carbohydrate and ketogenic diets [60]. However, these diets are almost impossible to maintain, and persons with severe depression do not have the strength needed to adhere to such demanding dietary regimes [61,62,63,64].

## 4. Materials and Methods

### 4.1. Subjects and Surgery

Intact male ICR mice from the Tel-Aviv University colony (Tel-Aviv, Israel), weighing 25–35 g, were used. The mice were maintained on a 12 h light: 12 h dark cycle with Purina rodent chow and water available ad libitum. Animals were housed in five per cage in a room maintained at 22 °C ± 0.5 °C until testing. Mice were used only once. The Faculty of Medicine Ethical Committee for Animal Experimentation approved all the experiments, which complied with the National Institutes of Health guidelines for animal experimentation of the [DHEW Publication (NIH) 85-23, revised, 1995].

### 4.2. Agents

Several agents were generously donated as follows: mianserin HCl was a generous gift from Rafa (Jerusalem, Israel); risperidone HCl was a generous gift from Abic (Netanya, Israel), amisulpride by Synthelabo Groupe (SylaChim, Mourenx, France), zolpidem by Unipharm (Tel-Aviv, Israel), mirtazapine was a generous gift from N.V. Organon (Oss, The Netherlands); venlafaxine HCl was a generous gift from Dexon (Hadera, Israel); reboxetine was purchased from Sigma-Aldrich Israel Ltd. (Rehovot, Israel). U50,488H {trans-3,4-dichloro-N-methyl-N-[2-(1-pyrrolidinyl)-cyclohexyl]-benzeneacetamide} by Upjohn Pharmaceutics (West Sussex, UK); naloxone HCl, nor-binaltorphamine (Nor-BNI) were obtained from the research technology branch of NIDA. All other compounds were procured from commercial sources. All the drugs were dissolved in saline except for amisulpride, dissolved in ethanol 70% and saline in 30:70 ratios. The administration of each drug followed the instructions of the company that supplied it: both in which medium to dissolve it and how to administer it (s.c. or i.p.). The drugs were prepared immediately before testing, and the requisite doses were given at a volume of 10 mL/kg. According to our previous studies [31,32,33,34,35,36,37], all agonists and antagonists were chosen. None of the antagonists possessed any analgesic effect, nor did they alter blood pressure. The agonist studied was used at sub-threshold doses and did not manifest any impact on their own.

### 4.3. Antinociception Assessment

#### 4.3.1. Hot Plate

Mice were tested with the hotplate analgesic meter Model 35D (IITC Inc., Woodland Hills, CA, USA), as previously described [31,32,33,34,35,36,37]. The device consisted of a metal plate (40 × 35 cm) heated to a constant temperature, on which a plastic cylinder was placed. The analgesic meter was set to a plate temperature of 52.5 °C ± 0.5 °C. The latency time was recorded between the second the animal was placed on the hotplate surface until it licked its back paw, jerked it strongly, or jumped out. Baseline latency was determined before experimental treatment for each mouse as the mean of two trials. Post-treatment latencies were determined after 30 min for all the drugs to minimize tissue damage, and a cut-off time of 30 s was adopted. Antinociception was defined quantitatively as a doubling of baseline values for each mouse. For each dose, at least ten different animals were checked, and their scores were summarized to determine the percentage of mice with analgesic responses. Each mouse was checked for only one treatment.

#### 4.3.2. Tail-Flick

Animals were tested for analgesia 30 min after drug injection using the tail-flick method, described in detail elsewhere [32,33]. Briefly, the baseline latencies were determined before experimental treatment for all animals as the means of two trials. The latency values were between 2.0 and 3.0 s. post-treatment latencies were determined as indicated for each experiment, and a maximal latency of 10 s was used to minimize tissue damage. Analgesia was defined quantitatively as a doubling or more baseline values for each mouse [31,32,33,34,35,36,37]. For each dose, at least ten different animals were checked, and their scores were summarized to determine the percentage of mice with analgesic responses. Each mouse was checked for only one treatment.

#### 4.3.3. Procedure

As described in the original studies published [31,32,33,34,35,36,37], all the antidepressant, antipsychotic, and hypnotic drugs used in the present study had opioid interaction. For each drug, the study was conducted over three different experiments.

#### 4.3.4. Experiment 1

Groups of mice (the exact number per group can be seen in Figure 1, Figure 2, Figure 3, Figure 4, Figure 5, Figure 6 and Figure 7) were injected with different doses, as indicated in the Results section, to determine the effect of each drug in eliciting antinociception. Normal motor behavior was observed following injection with no sedative effects. The doses of the drugs were chosen based on previous experience regarding the quasi-equipotent psychotropic drugs in acute pain animal models [31,32,33,34,35,36,37].

#### 4.3.5. Experiment 2

The sensitivity of each antidepressant to opioid antagonists was examined. First, we determined the effect of the nonselective opioid antagonist naloxone (1 or 10 mg/kg s.c.). In the next stage, we examine the sensitivity of each antidepressant to the k (kappa) opioid receptor antagonist Nor-BNI (10 mg/kg s.c.). In both stages, mice were injected simultaneously with the specific antidepressant drugs and the opioid antagonist.

#### 4.3.6. Experiment 3

The sensitivity of the antidepressant drugs to specific opioid agonists was examined as follows. Groups of mice (the exact number per group can be seen in Table 1) were given increasing doses of U50,488H, a selective kappa-receptor agonist; U50,488H was injected s.c. alone or with an inactive dose of the antidepressant drugs simultaneously.

### 4.4. Statistical Analysis

Quantitative dose–response curves were analyzed using an SPSS 18 computer program. This program maximized the log-likelihood function to fit a parallel set of Gaussian normal sigmoid curves to the dose–response data. The program calculated the dose–response curves and ED_50_ with 95% confidence limits. Single-dose antagonist studies were analyzed using the Fisher exact test. The analysis was performed on the binary antinociceptive response of the mice. In case of significant drug-by-dose interaction, post hoc analysis was performed using the Duncan method.

## 5. Conclusions

After an extensive assessment of the opioid interactions of many psychotropic medications with the opioid system in a mouse model of antinociception and the characterization of every one of them regarding agonistic and antagonistic interaction with the different opioid receptor subtypes, we hypothesize the possible use of those medications possessing antagonistic interactions with the kappa opioid receptors for the treatment of treatment-resistant depression (TRD), either as monotherapy or, better yet, as an add-on (augmentation) of the originally prescribed antidepressant. As our findings regarding the opioid involvement of the medications with the opioid system and specifically the k (kappa) opioid antagonistic effects of these seven drugs were all in an animal model, future clinical research should investigate the actual efficacy of this suggested approach. However, since these psychotropic medications are FDA-approved and largely prescribed for depression, the feasibility of conducting those needed studies is evident.

### 5.1. Limitations

As the results of many basic research studies have failed to be replicated when assessed in higher developmental animals and more so in humans, the translational process from the bench to the bedside (the clinic) is difficult, and, in the case of our studies (with rodents), needs to be first carried out in other large animals (i.e., pigs) and only then in human subjects. Another limitation is the fact that we have studied only several psychotropic medications, while there are many more antidepressants, antipsychotics, and sedative–hypnotic substances available for clinical use. Another limitation yet is the fact that the effect of Nor-BNI was tested 30 min post-injection. Following newer studies, a longer duration of time was suggested.

### 5.2. Strengths

All antidepressants, antipsychotics, and sedative–hypnotic medications were studied using the same strain of mice, and the same approach to the assessment of their antinociceptive effects in the model of nociception was taken. All medication doses were quasi-equipotent of those used in clinical settings, and all opioid agonist and antagonist dosages were used as already described in the literature. This enables us to compare the findings, which is much more complicated when, in most studies regarding other psychotropic medications, different rodents, and different models were used to characterize the opioid properties of the drugs.

## Figures and Tables

**Figure 1 pharmaceuticals-18-00208-f001:**
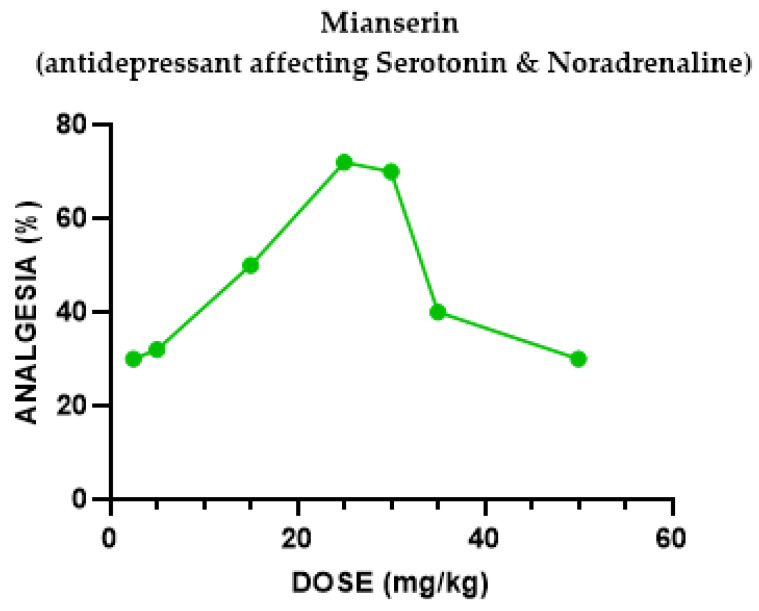
Dose–response curve, indicating the antinociceptive effect of Mianserin. Each group of mice (*n* = 10) received an i.p. injection and were tested with the analgesia hotplate meter test. Post-treatment latency was determined 60 min following injection.

**Figure 2 pharmaceuticals-18-00208-f002:**
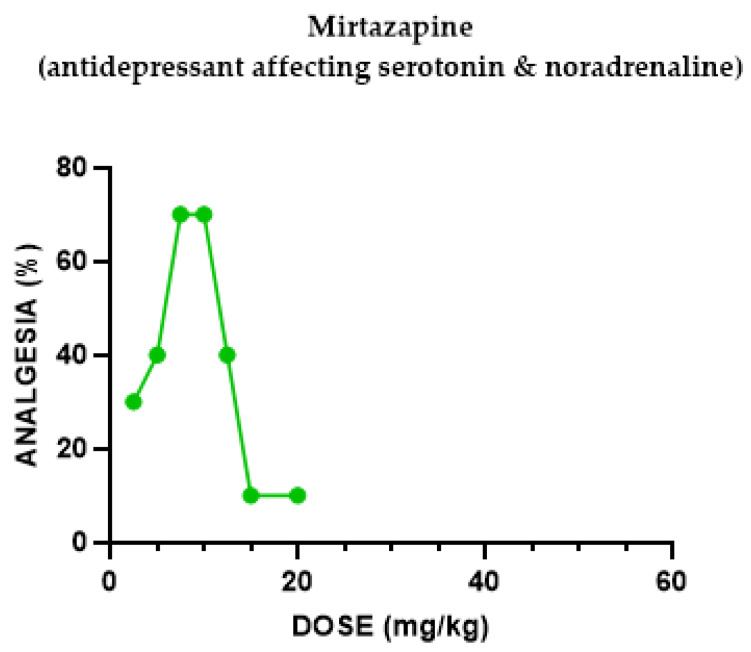
Dose–response curve indicating the antinociceptive effect of mirtazapine. Each group of mice (*n* = 20) received an i.p. injection and were tested with the analgesia hotplate meter test. Post-treatment latency was determined 60 min following injection.

**Figure 3 pharmaceuticals-18-00208-f003:**
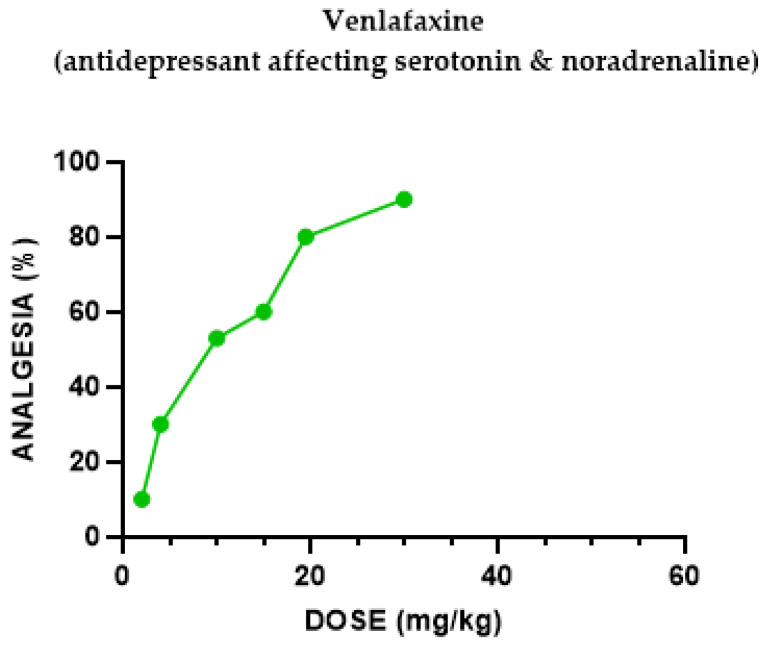
Dose–response curve indicating the antinociceptive effect of venlafaxine. Each group of mice (*n* = 10) received an i.p. injection and were tested with the analgesia hotplate meter test. Post-treatment latency was determined 30 min following injection.

**Figure 4 pharmaceuticals-18-00208-f004:**
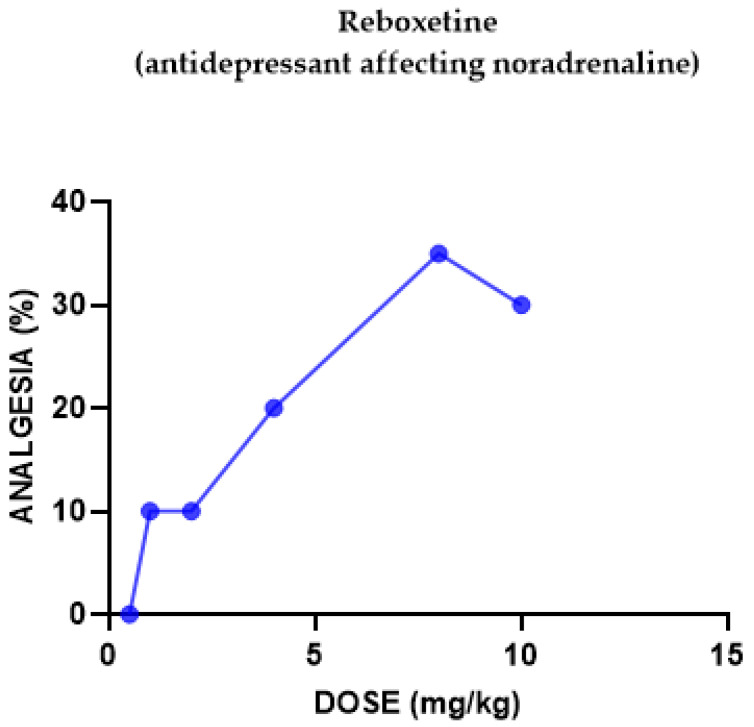
Dose–response curve indicating the antinociceptive effect of reboxetine. Each group of mice (*n* = 15) received an s.c. injection of reboxetine and was tested using the hotplate meter test. The post-treatment latency was determined after 30 min.

**Figure 5 pharmaceuticals-18-00208-f005:**
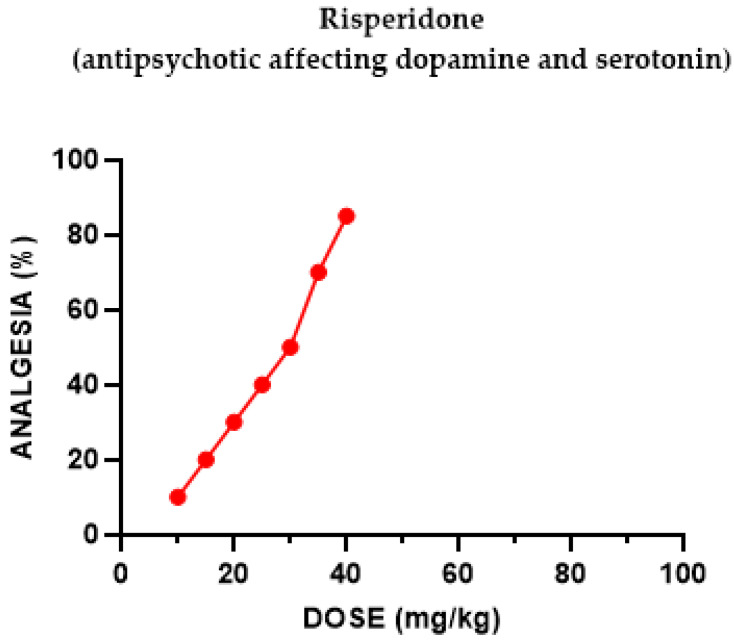
Dose–response curve indicating the antinociceptive effect of risperidone. Each group of mice (*n* = 20) received an i.p. injection and was tested using the tail-flick apparatus. The post-treatment latency was determined after 60 min.

**Figure 6 pharmaceuticals-18-00208-f006:**
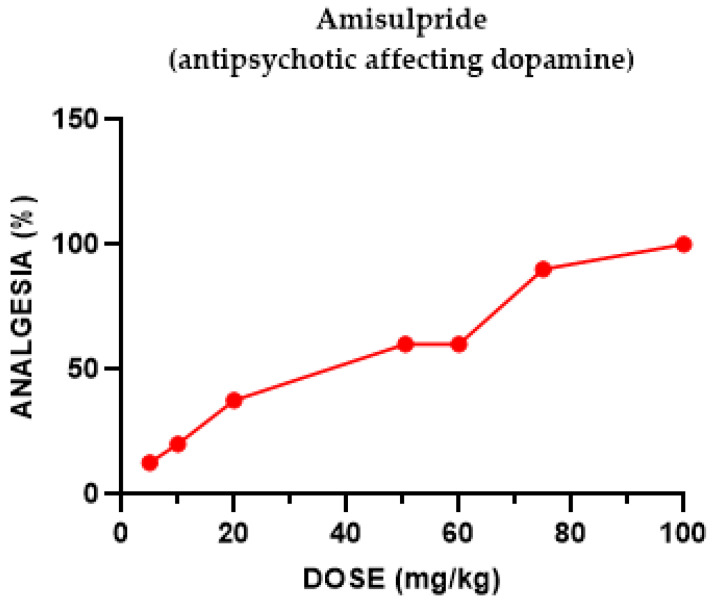
Dose–response curve indicating the antinociceptive effect of amisulpride. Each group of mice (*n* = 20) received an s.c. injection and was tested using the tail-flick apparatus. The post-treatment latency was determined after 30 min.

**Figure 7 pharmaceuticals-18-00208-f007:**
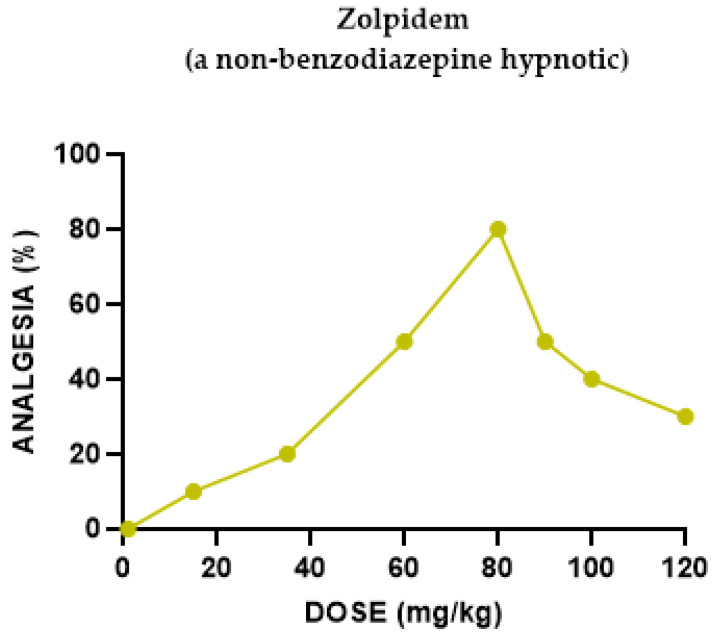
Dose–response curve indicating the antinociceptive effect of zolpidem. Each group of mice (*n* = 15) received an s.c. injection and was tested using the analgesia hotplate meter test 30 min later.

**Table 1 pharmaceuticals-18-00208-t001:** U50,488H interactions for all agents, either with or without an inactive dose. The numbers in parentheses are the 95% confidence limits of the ED_50_. * The asterisks indicate a significant difference from the group without U50,488H (*p* < 0.05). The N/A indicates interaction was not tested as it only showed a weak antinociceptive effect (mediated by nonselective opioid receptors) was found. In each group, n = 10, except amisulpride, which used *n* = 20 mice.

Agent	Without U50,488H	With U50,488H
Mianserin	4.8 (2.8; 10.3) mg/kg	0.5 (0.2; 1.2) mg/kg *
Mirtazapine	4.4 (1.9; 11.2) mg/kg	3.1 (1.3; 9.8) mg/kg
Venlafaxine	5.7 (2.2; 103.2) mg/kg	1.0 (0.4; 0.4) mg/kg *
Reboxetine	N/A	N/A
Risperidone	4.4 (1.9; 11.2) mg/kg	0.3 (0.1; 0.9) mg/kg *
Amisulpride	1.9 (0.09; 3) mg/kg	2.2 (0.2; 3.7) mg/kg
Zolpidem	Unchanged	Unchanged

**Table 2 pharmaceuticals-18-00208-t002:** The effects of the challenging of naloxone (1 mg/kg s.c. in all agents except zolpidem, Reboxetine 10 mg/kg) and Nor-BNI (10 mg/kg) opioid antagonists on all agents’ antinociceptive effect (Effect is the % of analgesia). Groups of mice ranged from risperidone, venlafaxine, and mianserin n = 10 through zolpidem and reboxetine n = 15 to amisulpride and mirtazapine n = 20. The asterisks indicate a significant decrease in analgesic response compared to the agent alone (*p* < 0.05).

Agent	Alone	Naloxone	Nor-BNI
Mianserin	70%	20% *	20% *
Mirtazapine	70%	10% *	20% *
Venlafaxine	80%	20% *	30% *
Reboxetine	40%	0% *	0% *
Risperidone	85%	20% *	15% *
Amisulpride	90%	30% *	15% *
Zolpidem	80%	30% *	20% *

## Data Availability

Data is contained within the article.

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
