# Peer review of "A New Trick of Old Dogs: Can Kappa Opioid Receptor Antagonist Properties of Antidepressants Assist in Treating Treatment-Resistant Depression (TRD)?"

_pharmaceuticals, 2025, doi:10.3390/ph18020208_

Round 1
Reviewer 1 Report
Comments and Suggestions for Authors
The objective of the present study was to examine the role of kappa opioid receptor mechanisms in the antinociceptive effects of various medications used in treating Major Depressive Disorder (MDD). The overall goal was to determine if kappa opioid receptor antagonism may be an effective form of therapy for those with treatment resistant depression. The antidepressants, antipsychotics, and sedative-hypnotic drugs examined showed antinociceptive (pain-relieving) effects in mice examined in the hot plate test, which were reversed by both naloxone and NorBNI. The authors suggest that kappa opioid receptor antagonism could be a promising mechanism in the treatment of treatment resistant depression. However, there are some significant concerns.
1) My main concern is the overall scope of what was examined in the study and the application of these findings to major depressive disorder and treatment-resistant depression. The study only examined the antinociceptive effects of these drugs using the hot plate and tail flick tests. Mice were not examined in any behavioral models of depression or negative affect. The results of the study only imply that the mechanisms these drugs act upon only regulate the antinociceptive properties of the drugs. Any conclusions regarding the potential of these mechanisms to aid in the treatment of treatment resistant depression are premature without testing them in models of depression.
2) The 30-minute pretreatment time of nor-BNI may not be sufficient to demonstrate a kappa opioid specific effect. Previous research has shown that nor-BNI acts a non-selective opioid receptor antagonist at shorter pretreatment times and that selective kappa opioid receptor antagonism in mice does not occur until 24 hours after initial treatment. Without longer pretreatment times where nor-BNI has been shown to selectively block kappa opioid receptors, conclusions cannot be drawn regarding the specific role of kappa opioid receptors in these tests.
Author Response
We sincerely appreciate your thoughtful remarks and valuable suggestions. We have carefully considered your feedback and made every effort to address your concerns.
The objective of the present study was to examine the role of kappa opioid receptor mechanisms in the antinociceptive effects of various medications used in treating Major Depressive Disorder (MDD). The overall goal was to determine if kappa opioid receptor antagonism may be an effective form of therapy for those with treatment resistant depression. The antidepressants, antipsychotics, and sedative-hypnotic drugs examined showed antinociceptive (pain-relieving) effects in mice examined in the hot plate test, which were reversed by both naloxone and NorBNI. The authors suggest that kappa opioid receptor antagonism could be a promising mechanism in the treatment of treatment resistant depression. However, there are some significant concerns.
Thank you for your comments. Just one important precision: The overall goal was not to determine if kappa opioid receptor antagonism may be an effective form of therapy for those with treatment-resistant depression but rather to suggest the possibility. We have changed the title to clarify it.
1) My main concern is the overall scope of what was examined in the study and the application of these findings to major depressive disorder and treatment-resistant depression. The study only examined the antinociceptive effects of these drugs using the hot plate and tail flick tests. Mice were not examined in any behavioral models of depression or negative affect. The results of the study only imply that the mechanisms these drugs act upon only regulate the antinociceptive properties of the drugs. Any conclusions regarding the potential of these mechanisms to aid in the treatment of treatment resistant depression are premature without testing them in models of depression.
We did not assess new compounds but rather “old” approved prescription medication (approved by the FDA for their indications, a process that follows phase I, II, and III studies). There was no need to examine the already-established efficacy of – let’s say the antidepressant mirtazapine (or venlafaxine) in a behavioral model of depression as its indication is the mere treatment of depression.
We suggest considering not just these medications’ main activity sites (post-synaptic serotonin and histamine receptors for mirtazapine, pre-synaptic reuptake inhibition of serotonin, noradrenaline, and, to a lesser degree, dopamine for venlafaxine) but also their interactions with the opioid system, particularly their antagonistic activity at the delta opioid receptor site.
2) The 30-minute pretreatment time of nor-BNI may not be sufficient to demonstrate a kappa opioid specific effect. Previous research has shown that nor-BNI acts a non-selective opioid receptor antagonist at shorter pretreatment times and that selective kappa opioid receptor antagonism in mice does not occur until 24 hours after initial treatment. Without longer pretreatment times where nor-BNI has been shown to selectively block kappa opioid receptors, conclusions cannot be drawn regarding the specific role of kappa opioid receptors in these tests.
As you can see in the manuscript, we have tested the opioid interactions of many psychotropic medications and found that only in a few of them did nor-BNI reverse its antinociception effect. Moreover, in several others, naloxone reversed the antinociception effect, but nor-BNI did not. This does not support a non-selective opioid receptor antagonistic effect of nor-BNI, at least in our studied models and with the same strain of mice we used in all the studies. We have added in the limitations section the following: “Another limitation yet is the fact that the effect of nor-BNI was tested 30 minutes post injection. Following newer studies, a longer duration of time was suggested”.
As already written, we suggest the possibility and do not conclude with a definite indication. The title raises the possibility, and the abstract ends with “Clinical studies are mandatory to establish the efficacy of the potential efficacy of augmentation of the treatment with antidepressants with these drugs in persons with treatment-resistant depression and the optimal dosage of medications prescribed.” (lines 36-38) We have now attenuated the phrasing of the conclusions to make it even clearer.
We addressed as many suggestions as possible. We thank the reviewers for their dedicated and thorough revision.
Reviewer 2 Report
Comments and Suggestions for Authors
Dear authors,
Thank you for your effort in contributing to the understanding of the relationship between kappa opioid receptors and the drugs used in the treatment of resistant depression.
The concept of your study is unique and has the potential to provide valuable insights into these aspects. However, the manuscript presented has significant shortcomings in the formulation of its objectives, the presentation of results, the discussion and comparison of findings, and it lacks a dedicated section summarizing the conclusions of your contributions.
I would therefore suggest a thorough revision, with a more detailed and structured approach that pays close attention to every detail. Below, I outline some of the most prominent issues identified in the submitted manuscript. A study like yours deserves to be supported by meticulous data handling and presentation, which are not currently reflected in the revieved document.
Comments:
Line 67: use TRD for treatment-resistant depression
Line 69: the term ‘eaketamine’ should be corrected
Figure 1: use lowercase letters for MG/KG (mg/kg)
It would be advisable to include both the number of animals and the dispersion data for the values used to generate the graph in the figure, either in the figure legend or within the main text. Additionally, a brief explanation of how the percentage of analgesia was calculated should be provided, which could be included in the Materials and Methods section.
Have the authors proposed any explanation for the observed U-shaped response? Furthermore, how would the authors account for a baseline analgesia of approximately 30%? This should be addressed and discussed in greater detail.
Line 212; Figure 4: Should be corrected “received an s.c injection and”
Line 215; 223: Should be corrected “the ED50”
The separation of the drugs for the graphical representation of their analgesic effects is unclear, in al figures. The figure captions should be standardized to align with the format used in Figure 1 for consistency.
Please provide an explanation for the use of one analgesia test for some animals/drugs and a different test for others. If there is a rationale for this approach, it should be clearly justified and substantiated.
Line 227: Throughout the manuscript, the authors should specify the exact number of animals or measurements conducted, as the use of the symbol "≥" for rporting the number is too imprecise. Additionally, when reporting percentage effects, the corresponding sample size (n) and variability data and graphical representation (e.g., SEM) should be included to provide a more accurate and reliable interpretation.
Line 235: It would be beneficial to present the results from this section in a quantitative table. Given that each drug may exhibit varying sensitivity to these changes, a self-explanatory table summarizing the data would help clarify these differences.
Line 242: This section also requires further clarification. For instance, the rationale behind the selection of mianserin should be provided. Additionally, an explanation of why it is referred to as the "inactive dose" should be included.
Have the potential sedative effects of all drugs used in this study been assessed? If so, how were these effects evaluated, and which techniques were employed? It is crucial to consider that pronounced sedative effects or impairments in motor coordination could significantly influence the observed antinociceptive outcomes. Furthermore, could co-administration of these drugs enhance their individual sedative or motor effects?
Table 1 should be self-explanatory. However, it lacks information regarding the meaning of the reported values. The units of measurement are not specified, nor is it clear which techniques were used to evaluate the effects presented.
Line 252-254: The statement, "These results suggest that mirtazapine, when administered together with κ opioid receptor subtypes, does not significantly change antinociception mechanisms," should be revised for clarity. This is a representative example of other similar claims made throughout this section. For instance, it would be more accurate to state that mirtazapine does not alter nociception. However, if the aim is to discuss the specific mechanisms involved with this drug, these should be more thoroughly explained and supported with appropriate evidence or literature.
Line 327: authors should homogenise terms such as Nor-BNI, nor-BNI or NorBNI. Similar for U50,488H, 50,488H and U50-488H; etc.
Discusion
The authors do not employ a model of acute pain per se, but rather techniques to quantify nociception or acute pain in healthy animals. This distinction should be clearly made throughout the manuscript to avoid any confusion.
Lines 279-328: In these lines of the discussion, the authors primarily summarize the results already presented in the corresponding section. However, they should incorporate a more in-depth analysis of these results, including comparisons with published studies. Additionally, the authors should provide possible explanations for the observed outcomes, such as the underlying mechanisms or interactions that could account for these findings.
From line 329 onwards, the discussion shifts to a more general exploration of treatments for treatment-resistant depression, the role of opioid receptors, and antipsychotics. However, no clear connection is made between these broader topics and the specific results obtained in the current study. The authors should establish a more direct link between their findings and the broader therapeutic context they discuss.
Author Response
We sincerely appreciate your thoughtful remarks and valuable suggestions. We have carefully considered your feedback and made every effort to address your concerns.
Comments and Suggestions for Authors
Dear authors,
Thank you for your effort in contributing to the understanding of the relationship between kappa opioid receptors and the drugs used in the treatment of resistant depression.
The concept of your study is unique and has the potential to provide valuable insights into these aspects. However, the manuscript presented has significant shortcomings in the formulation of its objectives, the presentation of results, the discussion and comparison of findings, and it lacks a dedicated section summarizing the conclusions of your contributions.
Thank you for this important suggestion: we have added in the discussion Tale no. 2 to clarify the results.
I would therefore suggest a thorough revision, with a more detailed and structured approach that pays close attention to every detail. Below, I outline some of the most prominent issues identified in the submitted manuscript. A study like yours deserves to be supported by meticulous data handling and presentation, which are not currently reflected in the revieved document.
Comments:
Line 67: use TRD for treatment-resistant depression
Changed as suggested. Thank you. (line 55)
Line 69: the term ‘eaketamine’ should be corrected
Corrected as suggested. Thank you. (line 71)
Figure 1: use lowercase letters for MG/KG (mg/kg)
Thank you, this has been corrected.
It would be advisable to include both the number of animals and the dispersion data for the values used to generate the graph in the figure, either in the figure legend or within the main text. Additionally, a brief explanation of how the percentage of analgesia was calculated should be provided, which could be included in the Materials and Methods section.
Have the authors proposed any explanation for the observed U-shaped response? Furthermore, how would the authors account for a baseline analgesia of approximately 30%? This should be addressed and discussed in greater detail.
A U-shape (or V-shape) response indicates a “therapeutic window” effect for the drug, a common phenomenon with psychotropic medications (e.g., lithium carbonate, clozapine, tricyclic antidepressants, and many anti-epileptics). We’ve added this now. Thank you. (lines 349-363)
Line 212; Figure 4: Should be corrected “received an s.c injection and”
Thank you, this has been corrected.
Line 215; 223: Should be corrected “the ED50”
Thank you, done as suggested.
The separation of the drugs for the graphical representation of their analgesic effects is unclear, in al figures. The figure captions should be standardized to align with the format used in Figure 1 for consistency.
Thank you, All the graphs have been changed following the request of reviewer no. 3. The new figure legends have been changed according to your request.
Please provide an explanation for the use of one analgesia test for some animals/drugs and a different test for others. If there is a rationale for this approach, it should be clearly justified and substantiated.
The tail-flick and the hot-plate antinociception tests are frequently used for the same purpose, and we are unaware of comparative studies demonstrating different findings between the tests while using the same protocol and the same mice strain when opioid interaction is assessed. Actually, when one apparatus was broken (and sent to repair), we used the other, and vice versa. Thank you.
Line 227: Throughout the manuscript, the authors should specify the exact number of animals or measurements conducted, as the use of the symbol "≥" for reporting the number is too imprecise. Additionally, when reporting percentage effects, the corresponding sample size (n) and variability data and graphical representation (e.g., SEM) should be included to provide a more accurate and reliable interpretation.
The exact number of animals was added to the figures and table. It was not changed in the discussion, as they are general remarks.
Line 235: It would be beneficial to present the results from this section in a quantitative table. Given that each drug may exhibit varying sensitivity to these changes, a self-explanatory table summarizing the data would help clarify these differences.
Thank you for this comment. A table (Table 2) was added to summarize this data.
Line 242: This section also requires further clarification. For instance, the rationale behind the selection of mianserin should be provided. Additionally, an explanation of why it is referred to as the "inactive dose" should be included.
We clarified in the paragraph starting in line 336 “In a series of previous studies, we assessed the antinociceptive effects of several psychotropic medications (antidepressants, antipsychotics, hypnotics) in a mouse model of acute pain. The aim was not to quantify the analgesic potency of the medications but to assess the possible interactions of these medications with the opioid system. We found several of them exerted antinociceptive effects involving the opioid system [24]. Others had no- or only weak opioid properties involved in their antinociceptive effects [31,34,35,37–40], while others yet had no antinociceptive effect at all [40–42].” Mianserin is one of the many antidepressants studied.
Have the potential sedative effects of all drugs used in this study been assessed? If so, how were these effects evaluated, and which techniques were employed? It is crucial to consider that pronounced sedative effects or impairments in motor coordination could significantly influence the observed antinociceptive outcomes. Furthermore, could co-administration of these drugs enhance their individual sedative or motor effects?
We did not assess the sedative or motor effects of the drugs. Each of these drugs has been thoroughly assessed during the phase-I, phase-II, and phase-III studies prior to their approval (by the FDA) for clinical use. As for the co-administration of these medications (not all of them at the same time, but various combinations) is a common clinical practice (i.e., adding an hypnotic or an antipsychotic medication to the treatment with an antidepressant).
Table 1 should be self-explanatory. However, it lacks information regarding the meaning of the reported values. The units of measurement are not specified, nor is it clear which techniques were used to evaluate the effects presented.
Thank you for this comment. Reported values were added, making Table 1 clearer.
Line 252-254: The statement, "These results suggest that mirtazapine, when administered together with κ opioid receptor subtypes, does not significantly change antinociception mechanisms," should be revised for clarity. This is a representative example of other similar claims made throughout this section. For instance, it would be more accurate to state that mirtazapine does not alter nociception. However, if the aim is to discuss the specific mechanisms involved with this drug, these should be more thoroughly explained and supported with appropriate evidence or literature.
Changed as suggested (not only for mirtazapine). Thank you.
Line 327: authors should homogenise terms such as Nor-BNI, nor-BNI or NorBNI. Similar for U50,488H, 50,488H and U50-488H; etc.
Homogenized as suggested (Nor-BNI; U50,488H). Thank you.
Discusion
The authors do not employ a model of acute pain per se, but rather techniques to quantify nociception or acute pain in healthy animals. This distinction should be clearly made throughout the manuscript to avoid any confusion.
In the introduction section, we have explicitly written: “Based on a large scale of basic studies (mouse models of antinociception) of the opioid interactions of various psychotropic medications (antidepressants, antipsychotics, hypnotics), we have suggested a possible beneficial effect of those antidepressants with agonistic interactions with the opioid system [24].” (lines 88-91).
We have not written anywhere that we are using a model of disease or disorder hence, it is implicit that healthy mice were used. In line 107, we’ve added the term “intact”
We’ve now added in the 1st paragraph of the discussion, “The aim was not to quantify the analgesic potency of the medications but to assess the possible interactions of these medications with the opioid system.” (line 338) Thank you.
Lines 279-328: In these lines of the discussion, the authors primarily summarize the results already presented in the corresponding section. However, they should incorporate a more in-depth analysis of these results, including comparisons with published studies. Additionally, the authors should provide possible explanations for the observed outcomes, such as the underlying mechanisms or interactions that could account for these findings.
This is a very delicate issue to address: Some Pharma companies tend, at times, to selectively publish data from studies conducted on their products (E.G., Ninan PT, Poole RM, Stiles GL. Selective publication of antidepressant trials. N Engl J Med. 2008 May 15;358(20):2181; author reply 2181-2; Schoones JW. Selective publication of antidepressant trials. N Engl J Med. 2008 May 15;358(20):2181; author reply 2181-2; de Jonge P, Bockting CL. Selective publication of antidepressant trials. N Engl J Med. 2008 May 15;358(20):2180-1; author reply 2181-2. doi: 10.1056/NEJMc080313), and the interactions of many psychotropic medications (used in psychiatry) with the opioid system are not common knowledge in psychiatric pharmacological literature. We are reluctant “to awaken demons from their slumber” and avoid discussing the issue as much as possible. We are sure you understand this, and thank you for that (too).
From line 329 onwards, the discussion shifts to a more general exploration of treatments for treatment-resistant depression, the role of opioid receptors, and antipsychotics. However, no clear connection is made between these broader topics and the specific results obtained in the current study. The authors should establish a more direct link between their findings and the broader therapeutic context they discuss.
Thank you for this suggestion. We have moved the short paragraph addressing the distribution of opioid receptors throughout the brain to the initial part of the discussion, allowing for a smooth transition to the history of opiates in the treatment of depression linking it with our findings. (line 330)
We addressed as many suggestions as possible. We thank the reviewers for their dedicated and thorough revision.
Reviewer 3 Report
Comments and Suggestions for Authors
This manuscript investigated the antinociceptive effects of several psychotropic medications and explores their potential kappa opioid receptor (KOR) antagonist properties, suggesting their possible use in treatment-resistant depression (TRD). While the study presented some interesting findings regarding the opioid system involvement in the antinociceptive effects of these medications, several significant weaknesses in the methodology and interpretation limit the strength of the conclusions drawn.
1. Main Research Question:
The main research question is whether some commonly used psychotropic medications possess KOR antagonist properties, and if so, whether this characteristic could contribute to their potential efficacy in TRD. While this was implied throughout the manuscript, it was not stated explicitly and concisely. A clearer articulation of the primary research question is needed in the introduction (e.g., line 40).
2. Originality and Relevance:
The manuscript builds on previous work by the same authors investigating opioid system interactions with psychotropic medications. The exploration of KOR antagonism in the context of TRD is timely and relevant, given the growing interest in KOR antagonists as potential antidepressants. The finding that several commonly used medications, including mianserin, mirtazapine, venlafaxine, and risperidone, show evidence of KOR antagonism in a mouse model, is potentially relevant to understanding their mechanisms of action and potential therapeutic applications. However, the reliance solely on antinociception in mice as a proxy for antidepressant activity severely limits the relevance to TRD in humans.
3. Related but Uncited Literature:
The manuscript overlooks several key references pertinent to the opioid system's role in depression and KOR antagonism’s therapeutic potential.
Some relevant references include:
- Carreno, F. R., & Frazer, A. (2017). Targeting the kappa opioid receptor for the treatment of mood disorders and other stress-induced pathologies. Frontiers in pharmacology, 8, 527.
- Al-Hasani, R., & Bruchas, M. R. (2011). Molecular mechanisms of opioid receptor-dependent signaling and behavior. Anesthesiology, 115(6), 1363-1381. (Provides more in-depth understanding of the complex opioid system)
- Knoll, A. T., & Carlezon Jr, W. A. (2010). Dynorphin, stress, and depression. Brain research, 1314, 56-73. (Focuses on the role of the dynorphin/KOR system in stress and depression, providing further context.)
- Crowley, N. A., Bloodgood, D. W., Hardaway, J. A., Kendra, A. M., McCall, J. G., Shaham, Y., & Bruchas, M. R. (2016). Dynorphin as a regulator of drug taking and seeking. Progress in neurobiology, 136, 22-43. (Addresses potential concerns about addictive liabilities, which were briefly mentioned in the manuscript.)
4. Methodological Improvements and Further Controls:
Several improvements are crucial for strengthening the study and increasing the reliability of the conclusions.
- Behavioral Test Specificity: Relying solely on the hotplate and tail-flick tests as the primary outcome measure is insufficient to justify implications for TRD. While antinociception can involve the opioid system, these tests do not specifically assess mood-related behaviours or address the complex neurobiological underpinnings of depression. Behavioral assays relevant to depressive-like symptoms, such as the forced swim test or sucrose preference test, should be included (line 128).
- Dose Selection: The manuscript's justification for the doses selected for psychotropic medications based on “quasi-equipotent psychotropic drugs in acute pain animal models” is unclear and requires more explanation. It is unclear how “quasi-equipotence” is defined or if the chosen doses translate to clinically relevant exposures in humans (line 160). Pharmacokinetic studies should ideally be conducted to ensure relevant doses are used and brain penetration is adequate.
- Kappa Agonist Experiments: The kappa agonist U50,488H experiments (line 168 and 240) should be more comprehensively designed to fully evaluate KOR interaction. Full dose-response curves with U50,488H in the presence and absence of the tested medications would provide stronger evidence of interaction than merely comparing ED50 values.
- Additional Controls: Several important controls are missing. It is not mentioned whether vehicles (used for dissolving the drugs) have been tested alone to exclude their effect on antinociception (line 121). The effects of stress caused by injection procedure were not evaluated by using sham-injected controls. Additionally, positive controls for KOR antagonism should be included to validate the assay (e.g., Nor-BNI alone at an established dose demonstrating a significant effect in both models)
- Blinding and Randomization: It is crucial to mention if any blinding and randomisation procedures were employed during the experiments to minimise bias (line 128).
5. Consistency of Conclusions with Evidence:
The conclusions, focusing on the potential role of KOR antagonism in TRD, significantly overreached the evidence presented. While the observed KOR antagonist properties of some medications were intriguing, the data from these acute pain models did not establish any direct relevance to the treatment of depression. The claim that the study’s results are "consistent with… observations on the possible antidepressant effect of substances exerting KOR antagonist properties" (line 90) requires significant substantiation, as the current findings lack behavioural tests directly linked to depression. The leap from antinociception to antidepressant effects is not supported by the current study design. It is misleading to link reduced acute pain responses with a clinically relevant treatment for chronic depressive states.
6. Tables and Figures:
- Figure 1's presentation of three separate dose-response curves on one graph with limited visual differentiation is problematic and could be enhanced by separating into three graphs or improving the distinguishability of the different datasets (line 191).
- Figure 2’s data showing only three doses of reboxetine provide limited evidence to characterise a weak antinociceptive effect, let alone involve kappa receptors, given that Naloxone is a global opiate receptor blocker. More doses for each curve for all of the drugs studied are required, especially reboxetine where only 3 dosages have been tested (line 209). More detailed testing at a wider range of doses is required.
- Table 1 needs better clarity. Reporting p-values next to statistically significant findings is mandatory (line 246), using descriptive statistical markers like asterisks (*) would also facilitate understanding. Explain N/A designation clearly and elaborate on whether an appropriate replacement control was established in instances with subthreshold doses that don't produce measurable effects by themselves (line 249).
7. General Caveats/Weaknesses/Mistakes:
- TRD Definition: While the manuscript provided a general description of TRD (line 54), the specific operational definition used to consider TRD within the study remains unstated.
- Animal Model Limitations: Using an acute pain model in mice to extrapolate findings to chronic human depression raises serious questions about the generalisability of results. The complex, multifactorial etiology of TRD is not captured by simply assessing antinociception, making it a very crude assessment.
- Mechanism of Antinociception: The observed KOR antagonist activity did not fully explain the overall mechanism of antinociception by these drugs. other systems are certainly also implicated in this phenomenon, for example, adrenergic mechanisms or effects on other types of opiate receptors. Future studies should dissect out the specific contribution of KOR antagonism relative to other pharmacodynamic effects. Any take on that?
- Overstated Clinical Implications: The frequent reference to TRD despite limited relevance of the methods to clinical scenarios should be toned down throughout the text (e.g., Lines 38, 287). Claims such as "Once… other medications are ineffective…" (Line 60), downplaying existing therapeutic strategies are also unjustified by presented data. Statements claiming superiority over keto diets or other treatment modalities for severe or TRD patients (lines 389-406) seem highly speculative given that no human subjects at all (let alone those following different diets or receiving those other treatments for TRD) have been assessed or even tested. These kinds of misleading assertions and premature assumptions compromise the scientific rigour of the manuscript.
- Drug Administration Route: Although stated explicitly and repeated, the varied route of drug administrations employed among experiments (some IP and other SC - line 225, for instance) without appropriate justification warrants caution in data interpretation across experiments and casts doubt about their possible additivity (line 105) given potential kinetic variability.
Author Response
We sincerely appreciate your thoughtful remarks and valuable suggestions. We have carefully considered your feedback and made every effort to address your concerns.
This manuscript investigated the antinociceptive effects of several psychotropic medications and explores their potential kappa opioid receptor (KOR) antagonist properties, suggesting their possible use in treatment-resistant depression (TRD). While the study presented some interesting findings regarding the opioid system involvement in the antinociceptive effects of these medications, several significant weaknesses in the methodology and interpretation limit the strength of the conclusions drawn.
- Main Research Question:
The main research question is whether some commonly used psychotropic medications possess KOR antagonist properties, and if so, whether this characteristic could contribute to their potential efficacy in TRD. While this was implied throughout the manuscript, it was not stated explicitly and concisely. A clearer articulation of the primary research question is needed in the introduction (e.g., line 40).
Thank you for this suggestion. We have added it in the introduction section. (line 86)
- Originality and Relevance:
The manuscript builds on previous work by the same authors investigating opioid system interactions with psychotropic medications. The exploration of KOR antagonism in the context of TRD is timely and relevant, given the growing interest in KOR antagonists as potential antidepressants. The finding that several commonly used medications, including mianserin, mirtazapine, venlafaxine, and risperidone, show evidence of KOR antagonism in a mouse model, is potentially relevant to understanding their mechanisms of action and potential therapeutic applications. However, the reliance solely on antinociception in mice as a proxy for antidepressant activity severely limits the relevance to TRD in humans.
The antidepressant activity of the antidepressant medications has been established years ago, in phase-I, phase-II, and phase-III studies before their approval by the FDA, and they are commonly prescribed for their indications. The findings regarding KOR antagonistic properties of some medications raise the possibility of studying their efficacy in clinical settings, as we’ve suggested in the abstract (line 36), the introduction (line 81), and at the end of the discussion (line 462). Thank you.
- Related but Uncited Literature:
The manuscript overlooks several key references pertinent to the opioid system's role in depression and KOR antagonism’s therapeutic potential.
Some relevant references include:
- Carreno, F. R., & Frazer, A. (2017). Targeting the kappa opioid receptor for the treatment of mood disorders and other stress-induced pathologies. Frontiers in pharmacology, 8, 527.
- Al-Hasani, R., & Bruchas, M. R. (2011). Molecular mechanisms of opioid receptor-dependent signaling and behavior. Anesthesiology, 115(6), 1363-1381. (Provides more in-depth understanding of the complex opioid system)
- Knoll, A. T., & Carlezon Jr, W. A. (2010). Dynorphin, stress, and depression. Brain research, 1314, 56-73. (Focuses on the role of the dynorphin/KOR system in stress and depression, providing further context.)
- Crowley, N. A., Bloodgood, D. W., Hardaway, J. A., Kendra, A. M., McCall, J. G., Shaham, Y., & Bruchas, M. R. (2016). Dynorphin as a regulator of drug taking and seeking. Progress in neurobiology, 136, 22-43. (Addresses potential concerns about addictive liabilities, which were briefly mentioned in the manuscript.)
Thank you for suggesting the references we’ve missed during the library search. They have been added to the manuscript now.
- Methodological Improvements and Further Controls:
Several improvements are crucial for strengthening the study and increasing the reliability of the conclusions.
- Behavioral Test Specificity:Relying solely on the hotplate and tail-flick tests as the primary outcome measure is insufficient to justify implications for TRD. While antinociception can involve the opioid system, these tests do not specifically assess mood-related behaviors or address the complex neurobiological underpinnings of depression. Behavioral assays relevant to depressive-like symptoms, such as the forced swim test or sucrose preference test, should be included (line 128).
We did not assess new compounds but rather “old” approved prescription medication (approved by the FDA for their indications, a process that follows phase I, II, and III studies). There was no need to examine the already-established efficacy of – let’s say the antidepressant mirtazapine (or venlafaxine) in a behavioral model of depression as its indication is the mere treatment of depression.
We suggest considering not just these medications’ main activity sites (post-synaptic serotonin and histamine receptors for mirtazapine, pre-synaptic reuptake inhibition of serotonin, noradrenaline, and, to a lesser degree, dopamine for venlafaxine) but also their interactions with the opioid system, particularly their antagonistic activity at the delta opioid receptor site.
- Dose Selection:The manuscript's justification for the doses selected for psychotropic medications based on “quasi-equipotent psychotropic drugs in acute pain animal models” is unclear and requires more explanation. It is unclear how “quasi-equipotence” is defined or if the chosen doses translate to clinically relevant exposures in humans (line 160). Pharmacokinetic studies should ideally be conducted to ensure relevant doses are used and brain penetration is adequate.
The quasi-equipotent doses were taken from studies of the pharma companies with their respective substances and roughly correlated with the dosages used clinically (in humans).
- Kappa Agonist Experiments:The kappa agonist U50,488H experiments (line 168 and 240) should be more comprehensively designed to fully evaluate KOR interaction. Full dose-response curves with U50,488H in the presence and absence of the tested medications would provide stronger evidence of interaction than merely comparing ED50 values.
- Additional Controls:Several important controls are missing. It is not mentioned whether vehicles (used for dissolving the drugs) have been tested alone to exclude their effect on antinociception (line 121). The effects of stress caused by injection procedure were not evaluated by using sham-injected controls. Additionally, positive controls for KOR antagonism should be included to validate the assay (e.g., Nor-BNI alone at an established dose demonstrating a significant effect in both models)
- Blinding and Randomization:It is crucial to mention if any blinding and randomisation procedures were employed during the experiments to minimise bias (line 128).
This present paper suggests a second look at specific findings from studies done and published originally years ago. We are suggesting a novel look at those findings and cannot change the original experiments, even though your remarks are valid.
- Consistency of Conclusions with Evidence:
The conclusions, focusing on the potential role of KOR antagonism in TRD, significantly overreached the evidence presented. While the observed KOR antagonist properties of some medications were intriguing, the data from these acute pain models did not establish any direct relevance to the treatment of depression. The claim that the study’s results are "consistent with… observations on the possible antidepressant effect of substances exerting KOR antagonist properties" (line 90) requires significant substantiation, as the current findings lack behavioural tests directly linked to depression. The leap from antinociception to antidepressant effects is not supported by the current study design. It is misleading to link reduced acute pain responses with a clinically relevant treatment for chronic depressive states.
We did not link reduced acute pain responses with treatment for chronic depressive states. We characterized the opioid interaction of several psychotropic medications and found some of them to exert antinociceptive properties antagonized by naloxone and, among them, few that exerted KOR antagonistic properties. We concluded the discussion: “As our findings regarding the opioid involvement of the medications with the opioid system and specifically the k (kappa) opioid antagonistic effects of these seven drugs were all in an animal model, future clinical research should investigate the actual efficacy of this suggested approach. However, since these psychotropic medications are FDA-approved and largely prescribed for depression, the feasibility of conducting those needed studies is evident.” (lines 468-474)
- Tables and Figures:
- Figure 1's presentation of three separate dose-response curves on one graph with limited visual differentiation is problematic and could be enhanced by separating into three graphs or improving the distinguishability of the different datasets (line 191).
Thank you. The graph in Fig. 1 has been split into 3 figures.
- Figure 2’s data showing only three doses of reboxetine provide limited evidence to characterise a weak antinociceptive effect, let alone involve kappa receptors, given that Naloxone is a global opiate receptor blocker. More doses for each curve for all of the drugs studied are required, especially reboxetine where only 3 dosages have been tested (line 209). More detailed testing at a wider range of doses is required.
Thank you. This present paper suggests a second look at specific findings from studies that were published years ago. We are suggesting a possible novel look at those findings and cannot change the original experiments, even though your remarks are valid. However, reboxetine was found to have a very weak antinociceptive effect.
- Table 1 needs better clarity. Reporting p-values next to statistically significant findings is mandatory (line 246), using descriptive statistical markers like asterisks (*) would also facilitate understanding. Explain N/A designation clearly and elaborate on whether an appropriate replacement control was established in instances with subthreshold doses that don't produce measurable effects by themselves (line 249).
We’ve added the asterisks, the mg/kg, and the p-value in Tale 1, as suggested. The N/A was explained in the figure legend: “The N/A indicates interaction not tested as only a weak antinociceptive effect (mediated by nonselective opioid receptors) was found.” (line 289). Thank you.
- General Caveats/Weaknesses/Mistakes:
- TRD Definition:While the manuscript provided a general description of TRD (line 54), the specific operational definition used to consider TRD within the study remains unstated.
Discussing the controversies regarding the exact definition of TRD is far beyond the scope of this paper. Sorry.
- Animal Model Limitations:Using an acute pain model in mice to extrapolate findings to chronic human depression raises serious questions about the generalisability of results. The complex, multifactorial etiology of TRD is not captured by simply assessing antinociception, making it a very crude assessment.
As explained earlier, we did not link reduced acute pain responses with treatment for chronic depressive states.
- Mechanism of Antinociception:The observed KOR antagonist activity did not fully explain the overall mechanism of antinociception by these drugs. other systems are certainly also implicated in this phenomenon, for example, adrenergic mechanisms or effects on other types of opiate receptors. Future studies should dissect out the specific contribution of KOR antagonism relative to other pharmacodynamic effects. Any take on that?
Our studies aimed to assess the possible interactions of several psychotropic medications with the opioid system. Not to study their potential efficacy in the treatment of pain. The role of (mainly spinal) noradrenaline and serotonin in pain mechanisms is well-known and not pertinent to this study.
- Overstated Clinical Implications:The frequent reference to TRD despite limited relevance of the methods to clinical scenarios should be toned down throughout the text (e.g., Lines 38, 287). Claims such as "Once… other medications are ineffective…" (Line 60), downplaying existing therapeutic strategies are also unjustified by presented data. Statements claiming superiority over keto diets or other treatment modalities for severe or TRD patients (lines 389-406) seem highly speculative given that no human subjects at all (let alone those following different diets or receiving those other treatments for TRD) have been assessed or even tested. These kinds of misleading assertions and premature assumptions compromise the scientific rigour of the manuscript.
We have attenuated as suggested and concluded, “Clinical studies are mandatory to establish the potential efficacy of augmentation of the treatment with antidepressants with these drugs in persons with treatment-resistant depression and the optimal dosage of medications prescribed.” (lines-36-38).
- Drug Administration Route:Although stated explicitly and repeated, the varied route of drug administrations employed among experiments (some IP and other SC - line 225, for instance) without appropriate justification warrants caution in data interpretation across experiments and casts doubt about their possible additivity (line 105) given potential kinetic variability.
The administration of each drug followed the instructions of the company that supplied it: both in which medium to dissolve it and how to administer it(s.c or i.p). An additional sentence was added to the agent’s part in the material and methods section. Thank you for this important remark. (lines 125-131)
We addressed as many suggestions as possible. We thank the reviewers for their dedicated and thorough revision.
Reviewer 4 Report
Comments and Suggestions for Authors
Authors provide an intriguing insight into the antinociceptive effects of psychotropic drugs and their sensitivity to opioid receptor antagonists. However, the lack of detailed quantitative analysis significantly limits the interpretation of findings and the ability to draw meaningful comparisons across the tested drugs.
To strengthen the impact and scientific rigor of this work, I recommend the following major revisions:
1. First, the study should include a detailed analysis of the degree of sensitivity of each drug to naloxone and NorBNI. This analysis should incorporate quantitative metrics, such as the percentage reversal of antinociceptive effects and statistical significance. If available, dose-response data for the antagonists should be presented to highlight differences in their potency or efficacy in reversing the effects of each psychotropic drug. Such analysis would provide valuable insights into the distinct mechanisms of action of these medications.
2. Second, the lack of a comparative efficacy assessment is a significant gap that needs addressing. Comparative evaluation is critical for translational relevance. Identifying which drugs exhibit superior antinociceptive effects and discussing how these findings align with their proposed mechanisms of action, particularly kappa opioid receptor involvement, would greatly enhance the study’s practical implications.
3. For drugs exhibiting biphasic dose-response curves, a detailed discussion of the potential mechanisms underlying these effects and their implications for therapeutic use is necessary. Biphasic responses suggest complex pharmacodynamics, which require careful explanation to contextualize the findings.
4. The introduction and discussion should include a more comprehensive examination of the multifaceted opioid system, particularly its overlapping receptor functions. Addressing this complexity is essential for ensuring the study’s design and interpretation support targeted and safe therapeutic outcomes.
5. If the authors adequately address the above concerns, the following 1-2 points may not need to be included in the study limitations; however, if these concerns remain unaddressed, all the points below should be clearly detailed in the limitations section.
1. The absence of direct comparisons or rankings of drug efficacy or potency. Although all drugs demonstrated sensitivity to opioid antagonists, the lack of clarity on which drugs are superior in producing antinociceptive effects undermines the study’s translational relevance.
2. The claim that all drugs are sensitive to naloxone and NorBNI is not substantiated with specific results or quantitative data. Detailed analysis of the degree of sensitivity and differences between drugs is necessary to improve transparency.
3. The study focuses on kappa opioid receptor antagonism but overlooks other potential pathways, such as serotonin or norepinephrine systems, which are particularly relevant for antidepressants.
4. The grouping of diverse psychotropic drugs (antidepressants, antipsychotics, and hypnotics) under a single paradigm does not account for their distinct pharmacological profiles, limiting the ability to draw class-specific conclusions or therapeutic implications.
5. While the findings are intriguing, their relevance to human physiology and clinical conditions remains speculative. Differences in receptor distribution and drug metabolism between mice and humans pose significant translational challenges.
6. The exclusive focus on antinociceptive effects, without considering broader behavioral or emotional impacts, restricts the understanding of the antidepressant or antipsychotic efficacy of these drugs in relevant contexts.
By addressing these points, the study will provide a more comprehensive and impactful analysis, enabling clearer conclusions regarding the role of opioid receptor systems in the antinociceptive effects of psychotropic medications. This will also help contextualize the findings for future therapeutic research.
Author Response
We sincerely appreciate your thoughtful remarks and valuable suggestions. We have carefully considered your feedback and made every effort to address your concerns.
Authors provide an intriguing insight into the antinociceptive effects of psychotropic drugs and their sensitivity to opioid receptor antagonists. However, the lack of detailed quantitative analysis significantly limits the interpretation of findings and the ability to draw meaningful comparisons across the tested drugs.
The quasi-equipotent doses were taken from studies of the pharma companies with their respective substances and roughly correlated with the dosages used clinically (in humans).
To strengthen the impact and scientific rigor of this work, I recommend the following major revisions:
- First, the study should include a detailed analysis of the degree of sensitivity of each drug to naloxone and NorBNI. This analysis should incorporate quantitative metrics, such as the percentage reversal of antinociceptive effects and statistical significance. If available, dose-response data for the antagonists should be presented to highlight differences in their potency or efficacy in reversing the effects of each psychotropic drug. Such analysis would provide valuable insights into the distinct mechanisms of action of these medications.
This manuscript focused on just one intriguing common finding relevant to some of the psychotropic medications studied (in the past). Detailed information is available for each medication in the original study publication (all of them are cited). This is a kind of “second look” at a specific finding regarding only some of the medications assessed in the past.
- Second, the lack of a comparative efficacy assessment is a significant gap that needs addressing. Comparative evaluation is critical for translational relevance. Identifying which drugs exhibit superior antinociceptive effects and discussing how these findings align with their proposed mechanisms of action, particularly kappa opioid receptor involvement, would greatly enhance the study’s practical implications.
The study has no clinical implications beyond the conclusion, “Clinical studies are mandatory to establish the potential efficacy of augmentation of the treatment with antidepressants with these drugs in persons with treatment-resistant depression and the optimal dosage of medications prescribed.”
- For drugs exhibiting biphasic dose-response curves, a detailed discussion of the potential mechanisms underlying these effects and their implications for therapeutic use is necessary. Biphasic responses suggest complex pharmacodynamics, which require careful explanation to contextualize the findings.
A U-shape (or V-shape) response indicates a “therapeutic window” effect for the drug, a common phenomenon with psychotropic medications (e.g., lithium carbonate, clozapine, tricyclic antidepressants, and many anti-epileptics). We’ve added this now. (lines 349-363). Thank you.
- The introduction and discussion should include a more comprehensive examination of the multifaceted opioid system, particularly its overlapping receptor functions. Addressing this complexity is essential for ensuring the study’s design and interpretation support targeted and safe therapeutic outcomes.
A comprehensive examination of the opioid system warrants a full chapter and is far beyond the scope of this paper. Sorry.
- If the authors adequately address the above concerns, the following 1-2 points may not need to be included in the study limitations; however, if these concerns remain unaddressed, all the points below should be clearly detailed in the limitations section.
- The absence of direct comparisons or rankings of drug efficacy or potency. Although all drugs demonstrated sensitivity to opioid antagonists, the lack of clarity on which drugs are superior in producing antinociceptive effects undermines the study’s translational relevance.
- The claim that all drugs are sensitive to naloxone and NorBNI is not substantiated with specific results or quantitative data. Detailed analysis of the degree of sensitivity and differences between drugs is necessary to improve transparency.
- The study focuses on kappa opioid receptor antagonism but overlooks other potential pathways, such as serotonin or norepinephrine systems, which are particularly relevant for antidepressants.
- The grouping of diverse psychotropic drugs (antidepressants, antipsychotics, and hypnotics) under a single paradigm does not account for their distinct pharmacological profiles, limiting the ability to draw class-specific conclusions or therapeutic implications.
- While the findings are intriguing, their relevance to human physiology and clinical conditions remains speculative. Differences in receptor distribution and drug metabolism between mice and humans pose significant translational challenges.
- The exclusive focus on antinociceptive effects, without considering broader behavioral or emotional impacts, restricts the understanding of the antidepressant or antipsychotic efficacy of these drugs in relevant contexts.
By addressing these points, the study will provide a more comprehensive and impactful analysis, enabling clearer conclusions regarding the role of opioid receptor systems in the antinociceptive effects of psychotropic medications. This will also help contextualize the findings for future therapeutic research.
We addressed as many suggestions as possible. We thank the reviewers for their dedicated and thorough revision.
Round 2
Reviewer 1 Report
Comments and Suggestions for Authors
I commend the authors on their work to improve the manuscript and have no further comments.
Author Response
We sincerely thank Reviewer 1 for your thoughtful remarks and positive feedback. We greatly appreciate your support of our work.
Reviewer 2 Report
Comments and Suggestions for Authors
Thank you very much for updating your manuscript. There are some formatting errors such as “Figure X” labels next to the figures that are redundant and the description of the tables are usually made at the top of them.
Author Response
Thank you for this remark. The figures have been changed following your suggestions.
Reviewer 4 Report
Comments and Suggestions for Authors
I appreciate the authors for considering the points raised and making the efforts to address some of the provided feedback. However, there are several areas where their responses do not fully address the concerns raised.
1. In relation to the quantitative analysis of sensitivity, authors refer to quasi-equipotent doses derived from previous studies but do not provide the requested quantitative analysis. While their emphasis on the study’s focus is noted, this response does not sufficiently address the core concern regarding the absence of specific data to substantiate claims about the sensitivity of the drugs to opioid antagonists. Including a summary of quantitative sensitivity results would enhance the study's credibility and provide greater clarity.
2. Regarding the comparative efficacy assessment, authors state that the study has no clinical implications beyond recommending further clinical studies. While this acknowledgment of a limitation is valid, it does not directly address the request for comparative efficacy data. Even within the stated scope, incorporating basic comparative analysis, such as ranking efficacy or potency, would add significant value to the findings.
3. On the topic of a comprehensive examination of the opioid system, the authors describe this as beyond the scope of the paper. While this position is understandable, a brief summary of the relevant complexities in the discussion would have provided helpful context for their findings, without needing a full chapter. The absence of such a summary limits the depth of the discussion.
4. The concern regarding the grouping of diverse drugs without class-specific analysis remains unaddressed. This omission reduces the study’s ability to derive meaningful conclusions about the class-specific effects or therapeutic implications of the psychotropic drugs studied.
5. With respect to transparency and data presentation, the authors rely on citations of original studies but do not provide specific data within the manuscript. Including a summary of key data from the cited studies would improve transparency and strengthen the manuscript’s presentation.
While the authors have addressed some feedback, such as providing an explanation of biphasic dose-response curves, their responses to key concerns—such as quantitative sensitivity analysis, comparative efficacy, and class-specific pharmacological profiles—remain insufficient. Addressing these issues would substantially improve the scientific rigor and impact of the study.
Author Response
Comments and Suggestions for Authors
I appreciate the authors for considering the points raised and making the efforts to address some of the provided feedback. However, there are several areas where their responses do not fully address the concerns raised.
- In relation to the quantitative analysis of sensitivity, authors refer to quasi-equipotent doses derived from previous studies but do not provide the requested quantitative analysis. While their emphasis on the study’s focus is noted, this response does not sufficiently address the core concern regarding the absence of specific data to substantiate claims about the sensitivity of the drugs to opioid antagonists. Including a summary of quantitative sensitivity results would enhance the study's credibility and provide greater clarity.
As mentioned in the letter to the editor (where we answered the reviewers’ comments), the quasi-equipotent doses were taken from studies of the pharma companies with their respective substances. Information was provided by the companies that sent us the substances, adding information on their solubility and the quasi-equipotent doses. It roughly correlated with the dosages used clinically (in humans). We did not perform quantitative analyses.
- Regarding the comparative efficacy assessment, authors state that the study has no clinical implications beyond recommending further clinical studies. While this acknowledgment of a limitation is valid, it does not directly address the request for comparative efficacy data. Even within the stated scope, incorporating basic comparative analysis, such as ranking efficacy or potency, would add significant value to the findings.
The comparative efficacy of psychotropic medication (antidepressants and antipsychotics) for their indications is roughly between 60% and 70%. Substances that do not reach this efficacy are not marketed, and unfortunately – there has been no new medication with a greater efficacy. The antinociceptive efficacy is not relevant to their suggested potential use for TRD.
- On the topic of a comprehensive examination of the opioid system, the authors describe this as beyond the scope of the paper. While this position is understandable, a brief summary of the relevant complexities in the discussion would have provided helpful context for their findings, without needing a full chapter. The absence of such a summary limits the depth of the discussion.
We understand the reviewer’s statement. However, we reiterate that it is far beyond the scope of this paper, as we do not want to transform it into a long review that most physicians treating TRD would find too long and too detailed to the point of becoming boring.
- The concern regarding the grouping of diverse drugs without class-specific analysis remains unaddressed. This omission reduces the study’s ability to derive meaningful conclusions about the class-specific effects or therapeutic implications of the psychotropic drugs studied.
Drugs are grouped according to the commonly accepted grouping in psychiatry. Unfortunately, unlike, for instance, the accepted grouping of antihypertension medications where drugs are grouped by their mechanism of action (i.e., beta-blockers, diuretics, calcium channel blockers, etc.), in psychiatry, this approach is not common (even though we do predict for it when teaching med. students both in the preclinical phase and during clerkship in psychiatry).
- With respect to transparency and data presentation, the authors rely on citations of original studies but do not provide specific data within the manuscript. Including a summary of key data from the cited studies would improve transparency and strengthen the manuscript’s presentation.
We understand the reviewer’s suggestion. However, as all data have already been published and are cited, we sustain that overloading this paper with detailed specific data (of the seven psychotropic medications found to exert opioid kappa-antagonistic effects, and more than the other 20 substances studied – some of them with various interactions with mu- delta- and kappa agonistic effects, others with no interaction with this system) would transform this paper into a too-long and too-boring paper to read. Furthermore, it would deviate the focus from the suggested hypothesis of using the described seven medications in the treatment of TRD. Sorry.
While the authors have addressed some feedback, such as providing an explanation of biphasic dose-response curves, their responses to key concerns—such as quantitative sensitivity analysis, comparative efficacy, and class-specific pharmacological profiles—remain insufficient. Addressing these issues would substantially improve the scientific rigor and impact of the study.
In the above answers, we have addressed those issues. Sorry for not fully satisfying the reviewer’s expectations.